

# Transcriptional profiling in the livers of rats after hypobaric hypoxia exposure

Zhenguo Xu[1,2,*], Zhilong Jia[1,2,*], Jinlong Shi[1,2], Zeyu Zhang[1,2], Xiaojian Gao[1,2], Qian Jia[2], Bohan Liu[2], Jixuan Liu[1,2], Chunlei Liu[1,2], Xiaojing Zhao[1,2] and Kunlun He[1,2]

[1] Laboratory of Translational Medicine, Chinese PLA General Hospital, Beijing, China
[2] Beijing Key Laboratory of Chronic Heart Failure Precision Medicine, Chinese PLA General Hospital, Beijing, China
[*] These authors contributed equally to this work.

## ABSTRACT

Ascent to high altitude feels uncomfortable in part because of a decreased partial pressure of oxygen due to the decrease in barometric pressure. The molecular mechanisms causing injury in liver tissue after exposure to a hypoxic environment are widely unknown. The liver must physiologically and metabolically change to improve tolerance to altitude-induced hypoxia. Since the liver is the largest metabolic organ and regulates many physiological and metabolic processes, it plays an important part in high altitude adaptation. The cellular response to hypoxia results in changes in the gene expression profile. The present study explores these changes in a rat model. To comprehensively investigate the gene expression and physiological changes under hypobaric hypoxia, we used genome-wide transcription profiling. Little is known about the genome-wide transcriptional response to acute and chronic hypobaric hypoxia in the livers of rats. In this study, we carried out RNA-Sequencing (RNA-Seq) of liver tissue from rats in three groups, normal control rats (L), rats exposed to acute hypobaric hypoxia for 2 weeks (W2L) and rats chronically exposed to hypobaric hypoxia for 4 weeks (W4L), to explore the transcriptional profile of acute and chronic mountain sickness in a mammal under a controlled time-course. We identified 497 differentially expressed genes between the three groups. A principal component analysis revealed large differences between the acute and chronic hypobaric hypoxia groups compared with the control group. Several immune-related and metabolic pathways, such as cytokine-cytokine receptor interaction and galactose metabolism, were highly enriched in the KEGG pathway analysis. Similar results were found in the Gene Ontology analysis. Cogena analysis showed that the immune-related pathways were mainly upregulated and enriched in the acute hypobaric hypoxia group.

# INTRODUCTION

The defining feature of a high altitude environment is sustained hypobaric hypoxic conditions. As terrestrial altitude increases, although the atmospheric proportion of oxygen remains constant at 21%, barometric pressure falls. In the meantime, the partial pressure of oxygen decreases at high altitude, giving rise to hypobaric hypoxia. For

Corresponding authors
Xiaojing Zhao, xjingzhao@126.com
Kunlun He, kunlunhe@plagh.org

humans ascending to a high altitude, the lower partial pressure of inspired oxygen leads to a reduction in the oxygen content of arterial blood, called systemic hypoxemia, and subsequently leads to tissue hypoxia with diminished cellular/mitochondrial oxygen availability (*Murray, 2016*). Accordingly, exposure to high altitudes may result in acute mountain sickness (AMS), a syndrome causing headaches and other symptoms occurring at altitudes >2,500 m (*Honigman et al., 1993*; *Roach et al., 2018*). Although it makes intuitive sense that arterial oxygen saturation should correlate with AMS symptoms, several studies have not found a significant association, at least at moderate altitudes similar to those at the Pace Laboratory at the Barcroft Station (3,800 m) (*Guo et al., 2014*; *O'Connor, Dubowitz & Bickler, 2004*). The majority of the world's population inhabits regions below 1,000 m elevation; nevertheless, our species has a remarkable capacity for hypoxia tolerance. Most notably, over tens of thousands of years, some high-altitude populations have adapted to life in this environment, with genetic signatures revealing natural selection around hypoxia-sensing pathways (*Bigham & Lee, 2014*). Approximately 140 million people live above 2,500 m, but approximately 40 million others venture into high altitude regions each year for work or leisure (*Weil, Glassner & Bosco 3rd, 2007*). These lowlanders experience a physiological response upon ascent, which has been well documented elsewhere and includes ventilatory, cardiovascular and erythropoietic factors. Even with adequate acclimatization, metabolic and physiological functions can be affected as the body reestablishes homeostasis under the hypoxic conditions. Energy metabolism is altered in heart and skeletal muscle. In the hypoxic mouse heart, decreased expression of proliferator-activated receptor alpha (PPARα) and its targets lowers fatty acid β-oxidation (FAO) capacity and represents a vital mechanism to conserve energetics and prevent hypoxic injury. In lowlander skeletal muscle, ATP and PCr levels fall at altitude and this loss continues over time. It is suggesting that the suppression of ATP supply is not met with a comparable down-regulation of ATP demand at altitude (*Hochachka et al., 1996*). In a study of Sherpa cardiac energy metabolism, carried out at sea level, a low cardiac phosphocreatine-to-ATP ratios (PCr/ATP) ratio was seen in comparison with lowlanders and this persisted even as the Sherpas acclimatized to sea level. Many other studies have also reported a downregulation of the expression and/or activity of FAO enzymes, many of which are PPARα targets, both in human muscle at altitude and in the heart and skeletal muscle of hypoxic rodents (*Horscroft et al., 2017*). This might indicate a failure to fully compensate for the lower oxygen availability or may itself form part of the acclimatization process, but it involves changes in gene expression and thus appears to be a regulated response. The positive aspects of high altitude acclimatization most notably decrease susceptibility to acute mountain sickness. However, the less well-understood aspect of high altitude deterioration is characterized by AMS symptoms including lethargy, fatigue and muscle wasting after prolonged exposure to extreme high altitude (>5,500 meter) (*Ward, 1954*). However, hypoxia is not only stress encountered at high altitude (*Weil, Glassner & Bosco 3rd, 2007*). Furthermore, hypoxia has been indicated to trigger vascular inflammation, which leads to increased vascular permeability, leukocyte adherence, and leukocyte emigration (*Jung et al., 2012*; *Lam et al., 2012*; *Wood et al., 2000*). Exposure to hypoxia promotes the expression of transcription factors, including nuclear factor NFκB (NF-κB), which plays a central role

in stimulating the release of proinflammatory cytokines (*Taylor, 2008*). Interleukin-1beta (IL-1β), IL-6, and tumor necrosis factor-α (TNF-α) have been suggested to increase under hypoxic conditions (*Dosek et al., 2007*; *Seys et al., 2013*). These proinflammatory cytokines are released by activated T cells and macrophages. However, in the literature (*Akopian et al., 2002*; *Bailey et al., 2004*), researchers have found that the elevation of these proinflammatory cytokines was not significant, which encouraged us to further study expression changes in tissues due to hypobaric hypoxia with the use of RNA-Seq analysis (*Wang et al., 2018*). Different tissues of the organism are required to diverse energy requirements under various ranges of oxygen concentrations (*Pompella & Corti, 2015*). As the largest metabolic organ in the body, the liver perform an important and complex biological functions under different oxygen concentrations. It has an important role in the metabolism of all kinds of component that are essential for survival, including carbohydrates, proteins, lipids etc. The hypoxia effect on different hepatocellular carcinoma cell lines have been analyzed before (*Bonewald, 1999*; *Ivanova et al., 2014*; *Kaelin Jr & Ratcliffe, 2008*). However, no in vivo studies have been conducted on the effect of hypobaric hypoxia exposure for 2 to 4 weeks on gene expression in the liver.

## MATERIALS & METHODS

### Animals and treatment

Adult Sprague-Dawley male rats were randomly allocated to 3 groups (4 animals per group) containing normal rats (L), rats exposed to hypobaric hypoxia for 2 weeks (W2L) and rats exposed to chronic hypobaric hypoxia for 4 weeks (W4L), as suggest in the article of *Ni et al. (2014)*. Rats in the W2L and W4L groups were exposed to a simulated altitude atmosphere with 5,500 m (380 mmHg), implemented by a FLYDWC50-1C low pressure hypoxic experimental cabin (Guizhou Fenglei Air Ordnance LTD, Guizhou, China). However, a rat in the acute exposure group died prior to the completion of the experiment. During breeding and experimental procedures, animals in both groups were housed in the same density per cage at a controlled ambient temperature of $25 \pm 2$ °C and $50 \pm 10\%$ relative humidity with a 12 h light/12 h dark cycle. Rats were given standard rodent chow and water ad libitum. Following overnight fasting, rats were sacrificed under anesthesia with 10% chloral hydrate (0.4 ml/100 g body weight, IP). The piece of right robe of liver was snap-frozen in liquid nitrogen and then stored at $-80$ °C until analysis anesthesia with 10% chloral hydrate (0.4 ml/100 g body weight, IP). Control rats (L) were anesthetized and sacrificed on day 1 and processed in the same manner as described above. Chinese PLA General Hospital Animal ethics committee provided full approval for this research (2017-X13-05).

### RNA quantification and qualification

RNA degradation and contamination were assessed on 1% agarose gels. RNA purity was checked using the NanoPhotometer® spectrophotometer (IMPLEN, Westlake Village, CA, USA). RNA concentration was measured using a Qubit® RNA Assay Kit in a Qubit® 2.0 Fluorometer (Life Technologies, Carlsbad, CA, USA). RNA integrity was assessed using

the RNA Nano 6000 Assay Kit from the Bioanalyzer 2100 system (Agilent Technologies, Santa Clara, CA, USA).

## Library preparation for liver Transcriptome sequencing

The input material for the RNA preparations used amount of 1 μg of RNA per sample. Following the manufacturer's recommendations, the sequencing libraries were generated with a NEBNext® UltraTM RNA Library Prep Kit for Illumina® (NEB, Ipswich, MA, USA) and the index codes were added to attribute sequences to per sample. Simply, The poly-T oligo-attached magnetic beads were used to mRNA purified from total RNA. Divalent cations were used in NEBNext First-Strand Synthesis Reaction Buffer (5X) to break at elevated temperatures. The first strand cDNA was synthesized using a random hexamer primer and M-MuLV reverse transcriptase (RNase H⁻). Subsequent use of DNA polymerase I and RNase H for second strand cDNA synthesis. The remaining overhangs are converted to blunt ends by exonuclease/polymerase activity. After adenylation at the 3′ end of the DNA fragment, NEBNext Adapters with a hairpin loop structure were ligated to prepare for hybridization. To preferentially select cDNA fragments of approximately 250–300 bp in length, library fragments were purified using the AMPure XP system (Beckman Coulter, Beverly, MA, USA).

Then, the size-ligated linker-ligated cDNA was used at 37 °C for 15 min and then at 95 °C for 5 min at 95 °C with 3 μl of USER Enzyme (NEB, Ipswich, MA, USA). Subsequently, PCR was carried out using Phusion High-Fidelity DNA polymerase, universal PCR primers and index (X) primers. Finally, the PCR product (AMPure XP system) was purified and library quality was assessed on an Agilent Bioanalyzer 2100 system.

## Clustering and sequencing

Indicated by the manufacturer's instructions, indexed samples was clustered on the cBot Cluster Generation System with the TruSeq PE Cluster Kit v3-cBot-HS (Illumina). Library preparations were sequenced on an Illumina Hiseq platform and generated 125 bp/150 bp paired-end reads after cluster generation.

## Data analysis
### Quality control

In this step, clean data is obtained by deleting reads containing adapters, readings containing ploy-N, and low quality reads from raw data. At the same time, the Q20, Q30 and GC contents of the cleaning data were calculated. All downstream analyses are based on high quality cleaning data. Raw fastq data were first processed by an internal Perl script. In this step, clean data were obtained by deleting reads containing adapters, reads containing a ploy-N and low quality reads from the raw reads. In the meantime time, the Q20, Q30 and GC content of the clean data were respectively calculated. The high-quality clean data were used in the downstream analyses.

### Mapping reads to the rat reference genome

Reference genome and gene annotation files were obtained directly from the genome website. The paired end clean readings were aligned to the reference genome with Hisat2

v 2.0.4. Hisat2 can generate a splice junctions database based on the gene annotation file and can obtain better mapping results than other non-splicing mapping tools.

### Differential expression analysis

The raw read counts of each sample were achieved using HTSeq v 0.9.1 (*Yu et al., 2012*). The normalized gene expression data were obtained using the calcNormFactors function from the edgeR package and voom function from the limma package. The L, W2L and W4L groups were compared using the limma package to obtain the differentially expressed genes (DEGs), filtered with the criteria that the corrected *p*-value was less than 0.05 and the minimum absolute log fold change of any three comparisons (W2L versus L, W4L versus W2L and W4L versus W2L) was more than 2.

### Functional analysis and gene set co-expression analysis of the DEGs

The clusterProfiler packages were used to implement the KEGG and Gene Ontology analysis of the DEGs (*Yu et al., 2012*). The default parameters were used in our analyses. Co-expression pathway analysis was done using the cogena package (*Jia et al., 2016*).

## RESULTS

### Differential expression analysis between the three groups

Clustering and principal component analysis of the samples based on the differentially expressed genes showed that responses to acute and chronic hypobaric hypoxia were different. A total of 497 DEGs were obtained via differential expression analysis (Table S1). A heat map of these genes between the three groups is presented with hierarchical clustering of the samples and genes in Fig. 1. Compared with the L control group, the expression patterns of differentially expressed genes in the W2L and W4L groups were largely different from each other, though they represented acute and chronic hypobaric hypoxia. As a result, the W4L and L groups were clustered more closely than the W2L and W4L groups, indicating a large difference between the W2L and W4L groups.

The principal component analysis of these samples using all the DEGs further verified this observation, as shown in Fig. 2. Briefly, the three groups were distinctly separated while samples in the same group were clustered, indicating that the rat models of acute and chronic hypobaric hypoxia were successful. The first principal component explained approximately 50% of the gene expression signatures and the W2L were highly separated from the L and W4L groups along the *x*-axis. Moreover, the W4L and L groups were distinct from each other in the second principal component, explaining approximately 20% of the variation.

### Immune-related pathways are highly enriched in a hypobaric hypoxia environment

To explore the biological function of the DEGs responding to hypobaric hypoxia, we used KEGG pathway and Gene Ontology enrichment analyses. Several innate immune pathways, including cytokine-cytokine receptor interaction, the IL-17 signaling pathway, the TNF signaling pathway and the chemokine signaling pathway, were highly enriched in the pathway analysis (Fig. 3). Additionally, some metabolic pathways, such as mineral

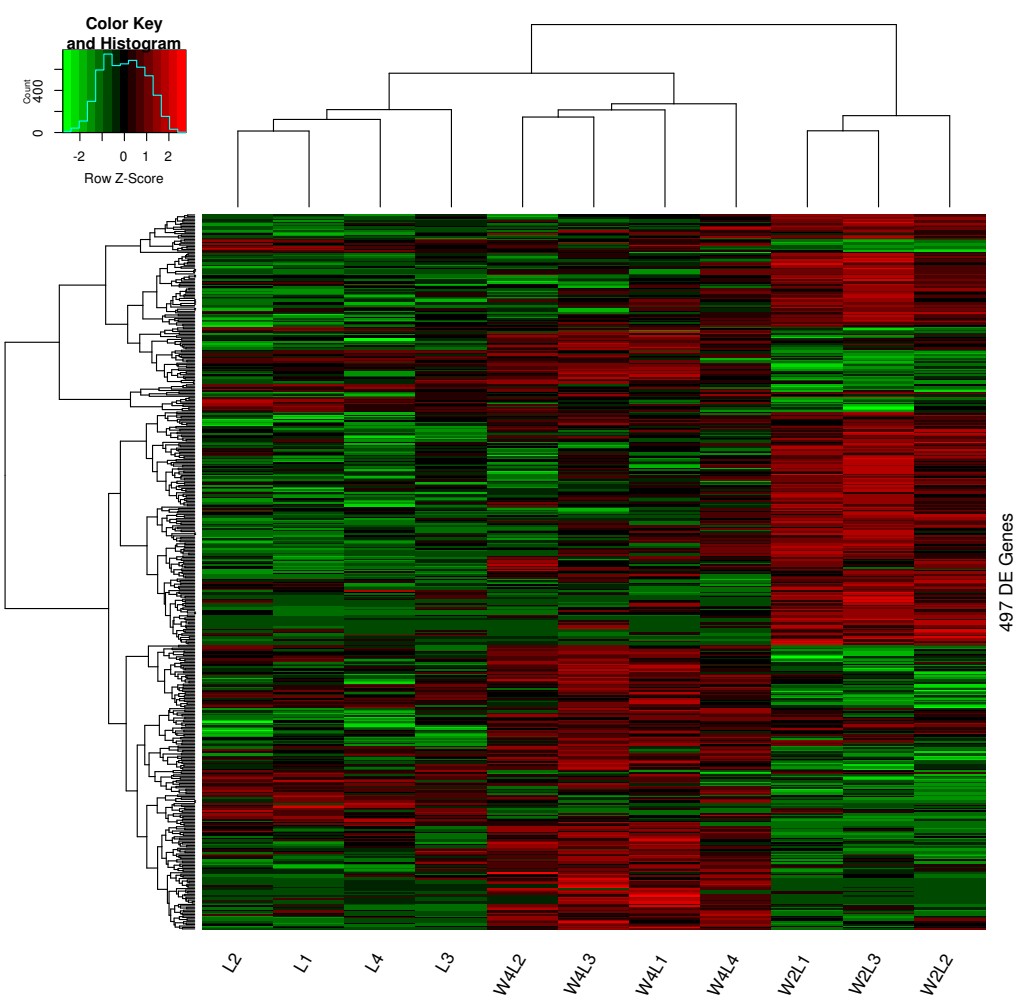

**Figure 1 Heatmap of the 497 DEGs in the three groups.** Colors represent the normalized gene expression values of DEGs. The control L group is separated with the hypobaric hypoxia groups. The W4L group is closer with the L group than with W2L group. The last number in the group labels represent sample id in this group.

absorption and galactose metabolism, were enriched as well. It seems that immune-related pathways represented in the DEGs are involved in the response and/or adaptation to hypobaric hypoxia.

We then used the Gene Ontology analysis to annotate the DEGs affected by hypobaric hypoxia (Fig. 4). Concerning molecular function, the GO analysis indicated that some migration and chemotaxis ontologies, such as leukocytes and granulocytes, were significantly enriched. For biological processes, receptor and ligand activity ontologies, especially cytokines and chemokines, were enriched. For cellular components, the GO analysis indicated that the DEGs were active in the extracellular matrix. The GO analysis verified the results of the KEGG pathway analysis.

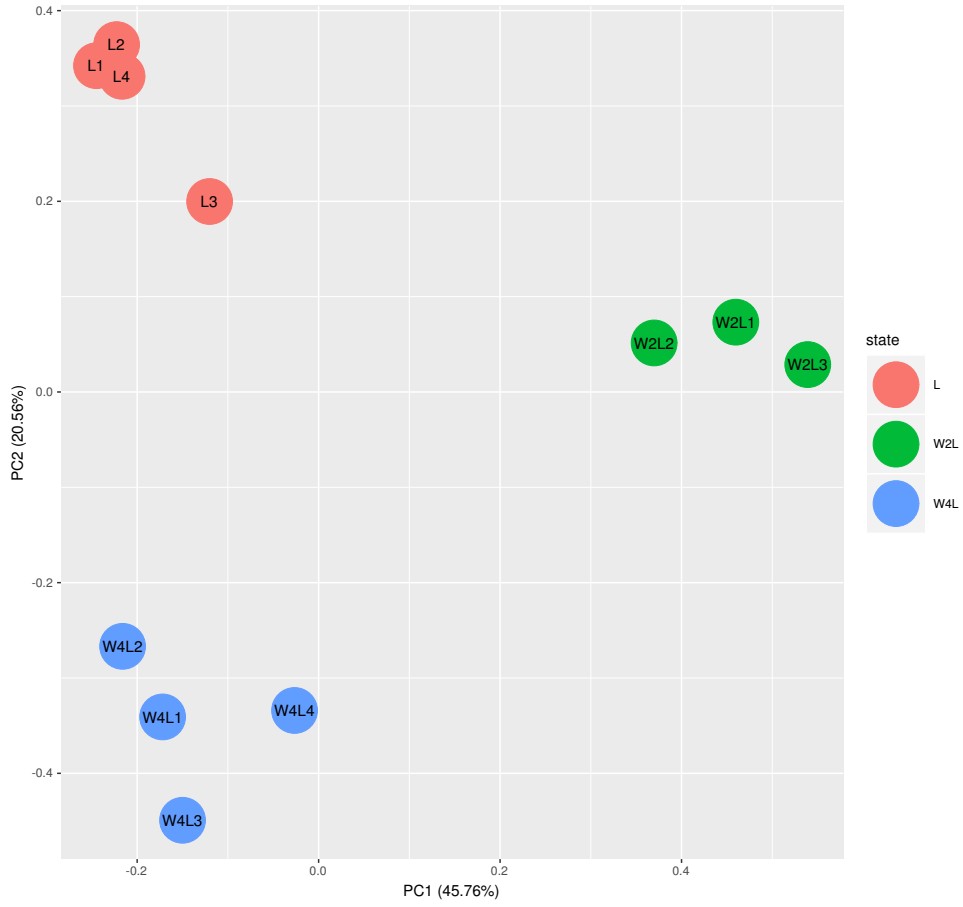

**Figure 2  Principal component analysis of the samples in the three groups.** The three groups are clearly separated.

## Co-expression pathway analysis

Three different co-expression patterns were observed in the co-expression analysis (Fig. 5). Co-expression analysis has the ability to cluster DEGs that possess the same or similar functions, which supplies a way to refine the DEGs and pathways. Most of the 244 genes in the first cluster were upregulated in week 2 but were downregulated in week 4 Genes in the second cluster, containing 148 genes, were upregulated until week 4.Most genes in the third cluster were downregulated in week 2 and then upregulated in week 4.

Pathway analysis for each co-expression cluster was done using the cogena bioconductor package, a tool for gene set co-expression analysis implemented by the authors. For the co-expression analysis of the DEGs, the k-means clustering method and three clusters for analysis were selected based on the cogena manual. Interestingly, immune-related pathways, such as the cytokine-cytokine receptor interaction pathway and the JAK-STAT signaling pathway, were enriched only in the first cluster but not in the other two clusters (Fig. 6). Combining this with the regulation direction of the genes in this cluster, it seems that immune-related pathways function in the first 2 weeks but not in week 4.

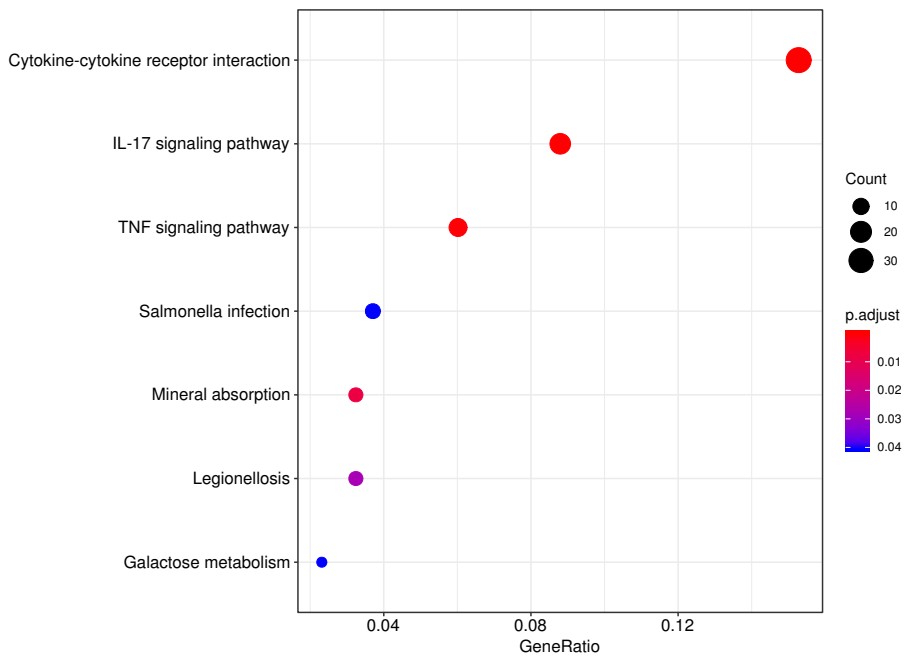

**Figure 3 Pathway analysis of the DEGs.** The top eight significant KEGG pathways are shown.

## DISCUSSION

This paper lays the first foundations for understanding transcriptional and signaling pathway changes in rat liver tissue under acute and chronic hypobaric hypoxia. These findings suggest that immune-related pathways play a key role in hypobaric hypoxia exposure and in the mechanistic differences between acute and chronic mountain sickness. The former study found an abundance of differentially expressed genes related to the immune system vary with an altitude of exposure to hypoxia (1,400 m, 3,000 m, 4,500 m). Unlikely, the immune system is not typically viewed as important under hypoxia in this study (*Baze, Schlauch & Hayes, 2010*). However, there is growing evidence that hypoxia may affect the immune system. Several studies of humans at high-altitude indicate perturbations of the immune system, especially about in circulating blood leukocytes. More studies suggest that hypoxia has a pro-inflammatory effect on macrophages, neutrophils and other white blood cells, and has anti-inflammatory effects on certain lymphocytes (*Bosco et al., 2006*; *Walmsley et al., 2005*). Furthermore, there is growing evidence that HIF is a key regulator of many immunological processes.

The present findings were obtained by comparing the gene expression profiles of liver tissue under high altitude conditions for 2 to 4 weeks with liver tissue under normal altitude conditions, providing clues to the molecular pathogenesis of this condition. Genome-wide transcriptional analysis suggests that hypoxia-induced proinflammatory cytokines and chemokines lead to liver injury in week 2. The study provides important information on the molecular mechanism causing liver injury at high altitudes and lays a foundation for subsequent gene validation and functional studies. An increasing number of articles in

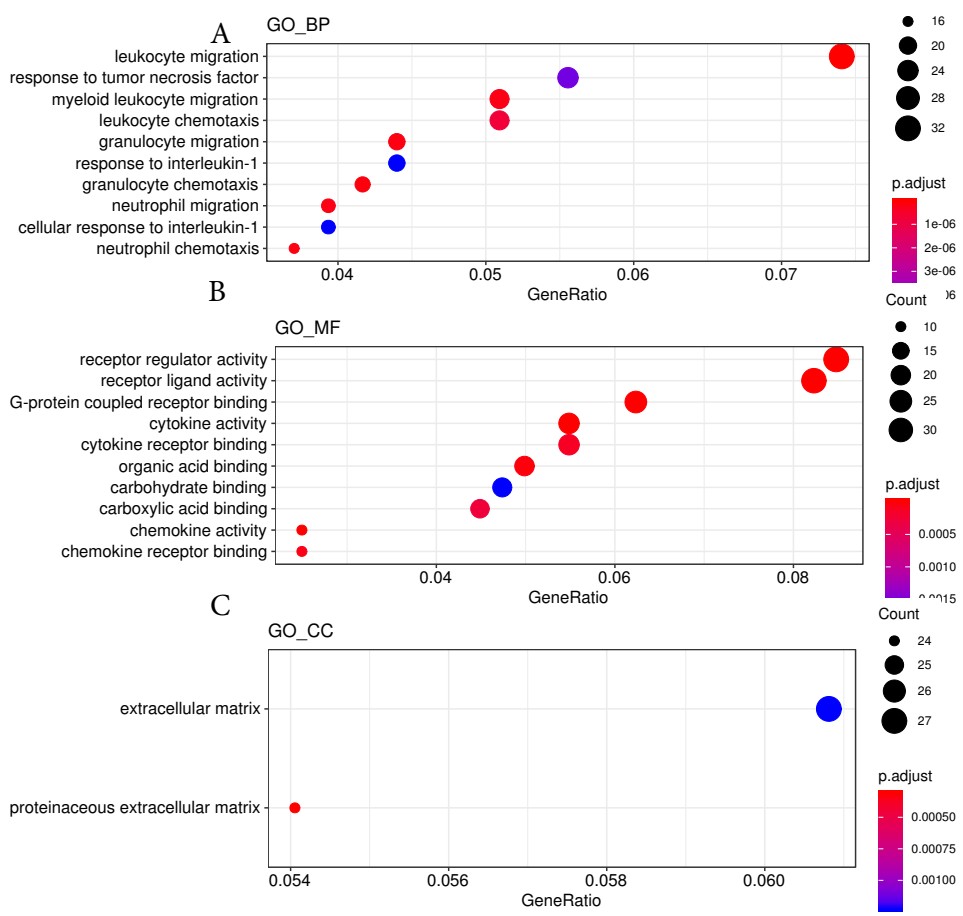

**Figure 4 GO analysis results from hypobaric hypoxia.** (A) The GO terms of molecular function (MF) were analyzed. (B) The GO terms of biological process (BP) were analyzed. (C) The GO terms of cellular component (CC)

the literature have indicated that a correlation between systematic inflammatory response and acute mountain sickness (AMS) exists. The reason may be that hypoxia was shown to alter cellular immunity and cytokine release. There is striking evidence that IL-6 may be involved in hypoxia-induced lung inflammation and pulmonary vascular remodeling and is possibly responsible for the occurrence of high-altitude diseases (*Savale et al., 2009*). Synthesis of IL-6 is stimulated by TNF and IL-1, which has been identified as a critical mediator of inflammation in tissues (*Li et al., 2015*).

Boos et al. reported that an increase in cytokines, such as IL-6 and IL-17α, was correlated with exercise at high altitudes(*Boos et al., 2016*). In addition, Lu et al. identified four cytokines including IGFBP6 (insulin-like growth factor binding protein 6), Dkk4 (dickkopf WNT signaling pathway inhibitor 4), SAA1 (serum amyloid A1), and IL-17RA (interleukin 17 receptor A), which might predict AMS susceptibility in a low-altitude environment (*Lu et al., 2016*). Liu et al. found IL-10 dysregulation, which is involved in immune and inflammatory responses, in AMS through transcriptome analysis. The reduction of IL-10

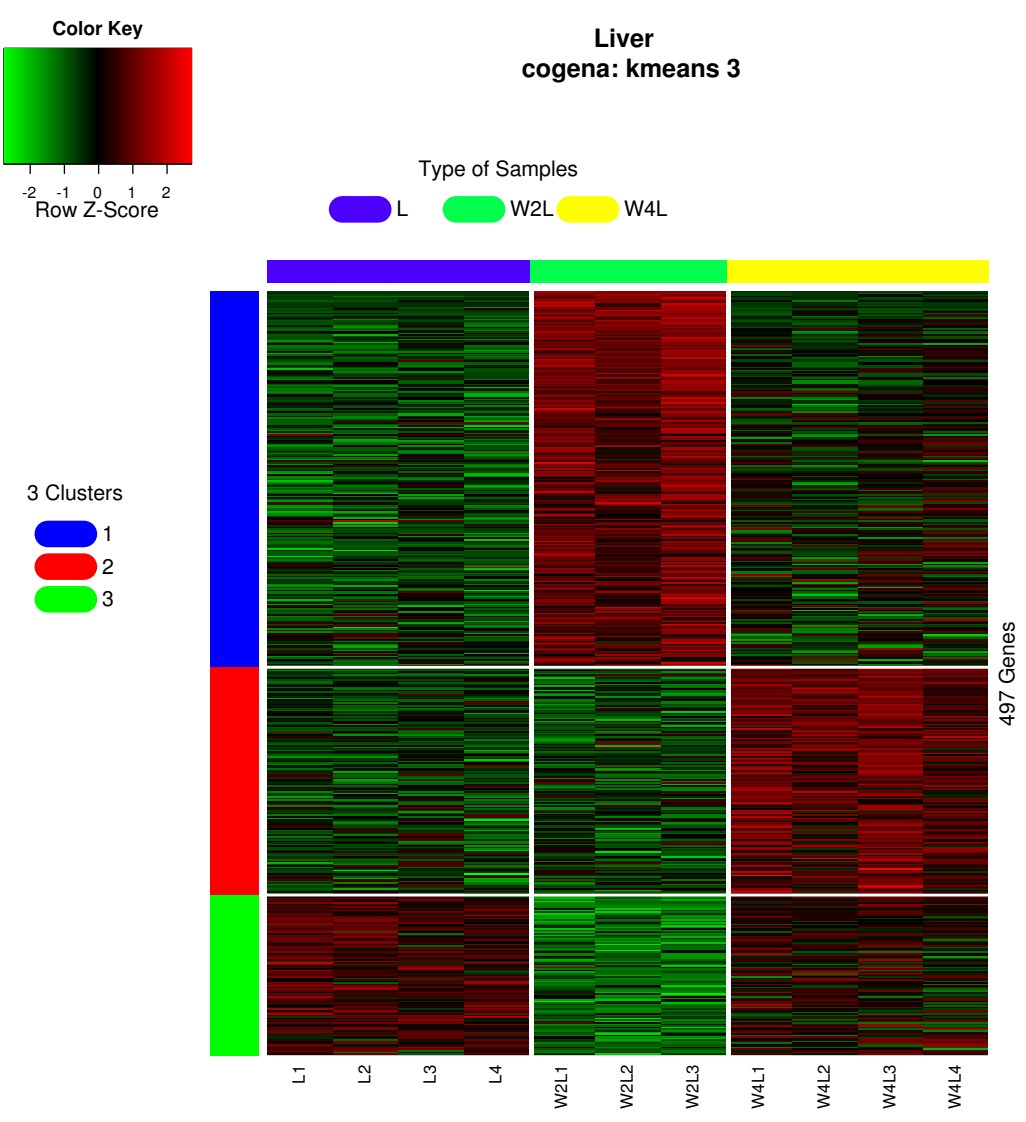

**Figure 5  Heatmap of co-expressed genes.** Kmeans clustering method and three clusters are shown in the heatmap, plotted with the cogena Bioconductor package.

after exposure to high altitude was strongly correlated with AMS (*Liu et al., 2017*). In addition to the proinflammatory cytokines, specific acute phase proteins have also been demonstrated to be changed under the condition of high altitude. Hypobaric hypoxia is a pathophysiological condition triggering the disturbance of cell organization, leading to protein, lipid, or DNA damage through oxidative acute mountain sickness stress (*Singh et al., 2010*).

In addition to causing symptoms of AMS, inflammation may play a key role in increasing ventilation, driven by increased chemosensitivity of the carotid body, which accompanies acclimatization to high altitude. A number of studies have documented increased cytokines and cytokine gene expression in the carotid body during exposure to acute or chronic
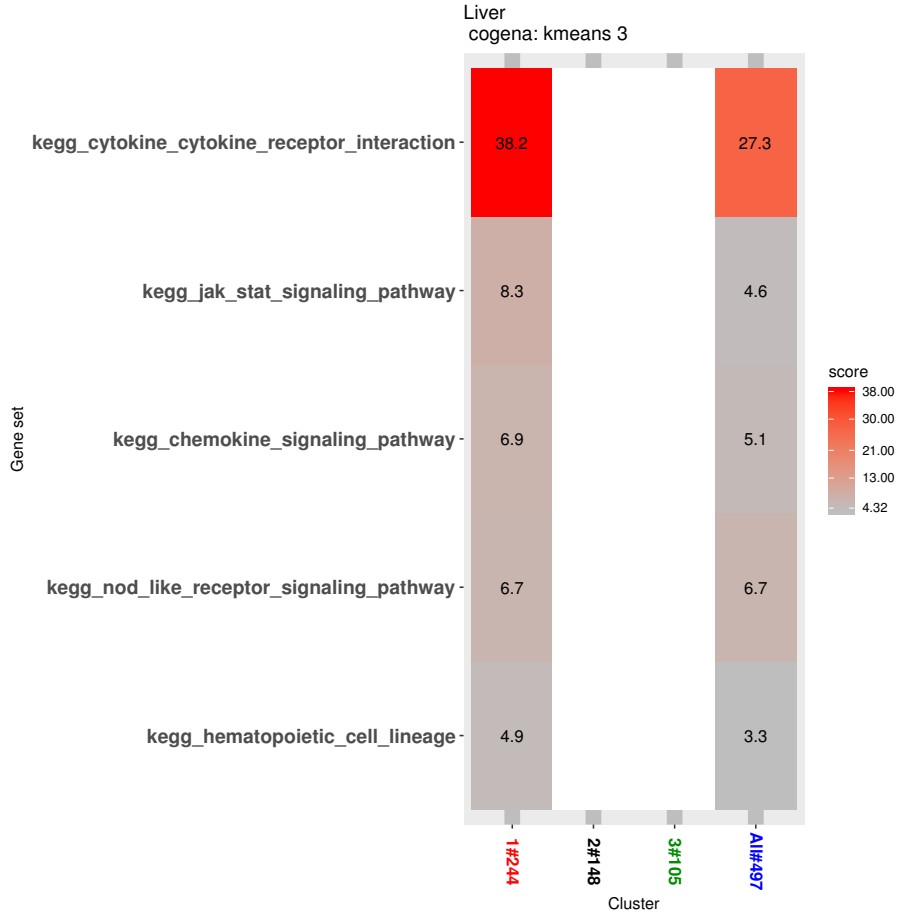

**Figure 6** **Co-expression pathway analysis.** KEGG pathways are shown with the enrichment score based on the cogena package.

hypoxia (*Lam et al., 2012*). Consistent with this, administration of anti-inflammatory drugs prevents increased cytokine expression and increased carotid body chemosensitivity in animals. A number of previous studies have documented increases in proinflammatory cytokines at high altitude, including IL-6 (*Klausen et al., 1997*). Furthermore, higher levels of interleukin receptor antagonist (IL-RA), a highly competitive antagonist of the proinflammatory cytokine IL-1, have been found in AMS-resistant subjects but not in AMS-sensitive subjects (*Julian et al., 2011*). However, there is no consensus on the role of inflammation in high-altitude acclimatization. Some argue that the elevation of cytokines occurs after the onset of AMS, and thus the time course of the inflammatory response contradicted the claim that inflammation plays a role in AMS (*Hartmann et al., 2000*). Furthermore, an older study shows unchanged concentrations of proinflammatory cytokines in response to hypoxia (*Swenson et al., 1997*). However, we find overexpression of the IL-17, TNF and chemokine signaling pathways in the liver under hypobaric hypoxia.

Notably, mineral absorption and galactose metabolish pathways were also over-represented. Concerning the galactose metabolism, most of the galactose enters the liver of

rat, where it is mainly converted to glucose, which is then either incorporated into glycogen or used for energy, connected with glycolysis. It seems that at the hypobaric hypoxia exposure, rats acquire more energy than as usual to survive. As far as we know, no study about the galactose metabolism in the hypobaric hypoxia condition was reported, though glycolysis was identified before (*Baze, Schlauch & Hayes, 2010*). The bone resorption and bone material properties are affected by the hypobaric hypoxia environment (*Bozzini et al., 2009*; *Guner et al., 2013*). Obviously, mineral absorption, induced by the hypobaric hypoxia, played critical roles in the bone resorption and bone mineral density. It indicates that the metabolism in rat is perturbed by the hypobaric hypoxia condition. These responses in metabolism are probably a way to adapt to the hypobaric hypoxia environment.

Pathway analysis for each co-expression cluster shows that the cytokine-cytokine receptor interaction pathway and JAK-STAT signaling pathway were enriched only in the first cluster. Dysregulation of the JAK -STAT pathways leads to hematopoietic and immune diseases. The JAK2-STAT pathways have been reported to be involved in cell protection and injury. The JAK2 inhibitor and overexpression of its dominant negative JAK2 protein improve cell survival against peroxide and superoxide anions. Inactivation of JAK2 has been shown to be a potential method in endothelial cells to avoid oxidative stress-induced death (*Neria et al., 2009*). Parthenolide has been reported to inhibit JAK1 and STAT3 activity. The product of reactive oxygen species (ROS) inhibits the STAT3 signaling pathway by targeting JAK1 (*Kurdi & Booz, 2007*).

Limitations of our study include the number of subjects, which could not generate precise cutoff values for AMS diagnosis in rats. In addition, we were not able to obtain blood samples in weeks 2 and 4; thus, we were unable to make a comparison and observe whether the changes in these indices are consistent with plasma.

## CONCLUSIONS

In conclusion, this study further verifies an increase in immune-related pathways in the first 2 weeks of high altitude exposure, but not in week 4. Our research found immune-related and metabolic pathways, such as cytokine-cytokine receptor interaction and galactose metabolism, were highly enriched in the KEGG pathway analysis for hypobaric hypoxia. Similar results were found in the Gene Ontology analysis. Cogena analysis showed that immune-related pathways were mainly upregulated and enriched for acute hypobaric hypoxia (2 week). It may be an important physiological cue in order for the rats to respond to a stressful environment. More studies are necessary to reveal the underlying molecular mechanisms in the liver controlling the expressions of key enzymes involved in the process.

### Funding

This study was supported by the Natural Science Foundation of China (31701155), the Chinese PLA General Hospital Medical large data project (No.2016MBD-002), the

Chinese PLA General Hospital Translational Medicine Project (No.2016-TM-013) and the Biological Medicine and Life Science Cultivation Foundation of Beijing Municipal Science and Technology Commission (No.Z151100003915075). The funders had no role in study design, data collection and analysis, decision to publish, or preparation of the manuscript.

### Grant Disclosures

The following grant information was disclosed by the authors:
Natural Science Foundation of China: 31701155.
Chinese PLA General Hospital Medical large data project: 2016MBD-002.
Chinese PLA General Hospital Translational Medicine Project: 2016-TM-013.
Biological Medicine and Life Science Cultivation Foundation of Beijing Municipal Science and Technology Commission: Z151100003915075.

### Competing Interests

The authors declare there are no competing interests.

### Author Contributions

- Zhenguo Xu conceived and designed the experiments, performed the experiments, authored or reviewed drafts of the paper, approved the final draft.
- Zhilong Jia analyzed the data, contributed reagents/materials/analysis tools, prepared figures and/or tables, authored or reviewed drafts of the paper, approved the final draft.
- Jinlong Shi analyzed the data.
- Zeyu Zhang and Chunlei Liu performed the experiments.
- Xiaojian Gao performed the experiments, authored or reviewed drafts of the paper.
- Qian Jia performed the experiments, prepared figures and/or tables, authored or reviewed drafts of the paper.
- Bohan Liu and Jixuan Liu contributed reagents/materials/analysis tools.
- Xiaojing Zhao conceived and designed the experiments, approved the final draft.
- Kunlun He conceived and designed the experiments, approved the final draft.

### Animal Ethics

The following information was supplied relating to ethical approvals (i.e., approving body and any reference numbers):

Chinese PLA General Hospital Animal ethics committee provided full approval for this purely observational research (2017-X13-05).

### Data Availability

The raw measurements are available in Table S1.

### Supplemental Information

Supplemental information for this article can be found online at http://dx.doi.org/10.7717/peerj.6499#supplemental-information.

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
