# Peer review of "Transcriptional profiling in the livers of rats after hypobaric hypoxia exposure"

_PeerJ, doi:10.7717/peerj.6499_

## Round 0.1 · original submission · Major Revisions

Please respond to all of the reviewer's comments

Reviewer 1 ·

Basic reporting

no comment

Experimental design

I have two questions for the authors regarding the experimental design of the current manuscript:

1) It wasn’t clear what were the conditions for the experimental groups, e.g. were they submitted to simulated altitude? What was the altitude tested? Any particular devices were used? Also, what are the genders of the rats used? Could there be gender difference?

2) It wasn’t clear to me how was the 2-week time point chosen. This is one of my major concern of the manuscript. Was any shorter period of hypobaric exposure tested? In my humble opinion acute effect should be measured at much shorter time points, such as 24-72 h. Would the authors explain their rational in choosing the 2-week time point? I would also like to see if the results would be different for a shorter period such as 48h.

Validity of the findings

Have the authors validated the results of the RNA-Seq?

Additional comments

In the manuscript "Transcriptional profiling in the livers of rats after hypobaric hypoxia exposure", the authors explored the gene expression in the livers of rats after 2 weeks or 4 weeks of hypobaric hypoxia exposure. In addition to my questions above, there are a few other things I would like to ask the authors:

1) In figure 6, it is interesting to see that all the pathways selected were enriched only in the first cluster, but no one showed up in the other two clusters. I am curious if any other pathways showed up in the other two clusters in this pathway analysis?
2) Since one of the major function of liver is metabolism, I am curious if the signal pathways/genes involved in metabolism are changed. The authors mentioned very briefly that the galactose metabolism pathway was enriched, but it seems no other information was provided or discussed.
3) Also, did the authors explore, or have plan to examine, the gene expression pattern in other organs under similar conditions?
4) There is a paper published on Physiol Genomics in 2010 titled “gene expression of the liver in response to chronic hypoxia” by Baze et al. (PMID 20103700), would the authors cite and discuss that paper?
5) Most of the figure legends are unreadable in the manuscript. Please correct them.
6) In the Abstract, the third sentence “The liver must physiologically and ….” and the fourth sentence “Since the liver is the largest metabolic organ…” should switch order.

Reviewer 2 ·

Basic reporting

1. In introduction, the language is not cohesive. Make sure to use connective words in between sentences to make introduction easier to read.

2. In introduction, a sentence starting at line 60 is very long. It fails to get the message across. I suggest authors break down that sentence into few small sentences to get the message across

Experimental design

1. In materials and method section, authors indicate that “the liver tissue was snap-frozen…”(line 92). What part of the liver was utilized for this study? Indicate if similar sections of the liver was isolated and utilized consistently throughout the study.

Validity of the findings

1. In introduction (line 57), authors suggest that energy metabolism of heart and skeletal muscle is altered. Based on the results of energy metabolism changes in liver, would there be extrapolation for anticipated changes (or literature evidence) in heart and skeletal muscle? Suggest include that in discussion.

2. One of the major finding for this study is that immune-related pathways play a key role in hypobaric hypoxia exposure. It would have been ideal to include certain functional end points in the study to have the evidence of translation of gene expression to functional end points. However, can authors suggest the evidence of the functional end points in literature in discussion to support major finding?

---

## Round 0.2 · accepted · Accept

Thank you to contribute to this interesting manuscript to PeerJ and we are happy to work with to publish this work.

# Reviewer 2 ·

Basic reporting

No further comments.

Experimental design

No further comments.

Validity of the findings

No further comments.

Additional comments

Authors have addressed my concerns. No further comments.